# Government as a Facilitator versus Inhibitor of Social Entrepreneurship in Times of Public Health Emergencies

**DOI:** 10.3390/ijerph20065071

**Published:** 2023-03-13

**Authors:** Shah Muhammad Kamran, Abdelmohsen A. Nassani, Muhammad Moinuddin Qazi Abro, Mahvish Kanwal Khaskhely, Mohamed Haffar

**Affiliations:** 1Institute of Science Technology and Development, Mehran University of Engineering and Technology, Jamshoro 76062, Pakistan; 2Department of Management, College of Business Administration, King Saud University, P.O. Box 71115, Riyadh 11587, Saudi Arabia; 3Department of Management, Birmingham Business School, University of Birmingham, Edgbaston, Birmingham B15 2TT, UK

**Keywords:** COVID-19, public health emergency, social entrepreneurship, role of government, health governance

## Abstract

COVID-19 established the need for even more social entrepreneurship globally. It is important for keeping society together in times of crises because it creates an environment that improves the quality of life during hard times and public health emergencies such as COVID-19. Even though it plays a unique role in returning things back to normal after a crisis, it faces opposition from many parts of society, especially the government. Still, there are not many studies that look at what the government should do to help or stop social enterprise during public health emergencies. That is why the goal of this study was to find out how the government has helped or hindered social entrepreneurs. Content analysis was conducted on the carefully mined data from the internet. The research found that regulations for social enterprises should be loosened, especially during and after pandemics and disasters. This could also make it easier to accomplish things in the government. It was also found that, in addition to financial help, capacity building through training can help social enterprises do more and make a bigger difference. This research provides broader guidelines for policymakers and new entrants in the field.

## 1. Introduction

The emergence of COVID-19 in Wuhan, China, as a public health emergency, has exposed the world to multiple challenges, ranging from health and the economy to poverty and hunger. As the global economy was in a shutdown, households dependent on daily wages and self-employed individuals, primarily labourers and low-level employees of SMEs, were hit the hardest, particularly in low- and middle-income countries. To handle this public health emergency and its negative consequences, governments around the world allocated special stimulus packages [1,2,3,4]. These packages were made by governments to assist their citizens who were experiencing financial distress.

This public health emergency targeted every nation on the planet, and G20 nations were as hard hit by the pandemic as small economies. Governments around the globe have issued financial packages under the categories of relief, support, and stimulus, among others, which supported their public in turmoil and social distancing. This pandemic has made it harder for governments to keep running smoothly, and it has made it harder for social entrepreneurs to get their products and services to people in need. Social entrepreneurs’ primary purpose is to make a social impact, especially when needed the most, e.g., in disaster and pandemics. Social entrepreneurs deliver goods and services in the market in an innovative manner that not only generates profit for owners and shareholders but also fulfils the needs of time and society. Like any other firm, social enterprises involve consumers, employees, and stakeholders, affected by their commercial endeavours [5]. Social entrepreneurs and enterprises also help governments by reaching out to remote areas and by developing innovative solutions to unresolved community and societal issues [6]. This means that they are needed to overcome challenges resulting from a crisis by providing strategic leadership.

The public health emergency because of COVID-19 created special circumstances as well as a set of opportunities for social enterprises. Commercial firms and enterprises are not compelled to perform their duties for only the greater social good. Nonetheless, social enterprises have a great deal to do for the common good of humanity due to their social nature, which requires them to step up and share the burden of government during times of crises. The age of social alienation has pushed entrepreneurs to come up with creative and workable solutions. Social entrepreneurs must use social innovation for the common good to bridge the gap.

According to studies, the public sector, or social entrepreneurship, is in the process of creating value for citizens by combining private and public resources to explore social opportunities. Further, COVID-19 has brought colossal challenges for governments on the economic front, including strain on medical services and care and the need for social support to meet the public’s basic needs. In this regard, the government’s limited resources and capabilities to respond to this pandemic by managing its support services provide an opportunity for social entrepreneurs among other partnership agents, who support vulnerable populations [7,8,9,10].

Moreover, in many economies, political decision making, which is attributed to government, plays a crucial role in the development and motivation of entrepreneurship through defining policies and establishing infrastructure inclusive of support and network mechanisms [11,12]. Conversely, variables such as different taxes and bureaucracy in the government machinery can break the entrepreneurial spirit in society [13]. From this discussion, it can be inferred that both government and social entrepreneurship work together in the successful management of pandemic and disasters and any other kind of public health emergency. Further, despite the repercussions caused by the COVID-19 scenario, the possibilities of positive spill overs have emerged in terms of social policy initiatives. This means that the government is a part of the social entrepreneurial process as asserted by the “emergence of social enterprise in Europe school of thought”. According to this, the government plays a critical role in fostering and/or inhibiting social entrepreneurship through rules and regulations related to social value creation. The main type of social enterprise initiatives are knowledge development or public awareness and service or product development and delivery, which can alleviate social problems derived from COVID-19 [14].

### 1.1. Problem Statement

Therefore, the role of government in affecting the functioning of social entrepreneurs is important for the effective management of the damage caused by this exogenous event (COVID-19). Further, exploring the government’s part in both the facilitation and inhibition of social entrepreneurship can guide the drafting of effective policies and interventions to enhance the former role while reducing bureaucratic and other barriers associated with the government machinery. Many studies, including Kamran et al. (2022), Bacq et al. (2020), and others [6,15], have investigated the role of social entrepreneurship in assisting the government during times of social distancing. However, the role of government in influencing social entrepreneurship ventures during the COVID-19 pandemic has not been sufficiently addressed in the scholarly literature, as asserted by many studies [16,17,18,19,20]. As a result, the current study aims to fill this research gap by identifying the government’s barriers and supporting factors for social entrepreneurs in addressing the crisis associated with the COVID-19 pandemic. The precise research question is “How are governments facilitating or potentially inhibiting the emergence of a robust social enterprise sector, specifically in times of extremes such as the COVID-19 pandemic”? To attain the purpose of this research, a context-responsiveness framework for the relationship between government and social entrepreneurship is employed [21] since we explore the cases of social entrepreneurship endeavours from the perspective of developing as well as developed countries.

### 1.2. Theoretical and Practical Contribution of the Study

More research is needed to develop policy recommendations for government support in social enterprises that enable innovative e-Government service delivery, according to a recent World Bank policy directive [22]. It shows the importance of the economy and society as a whole for government and social entrepreneurs to work together. Therefore, this research contributes to this fast-evolving stream of social entrepreneurship literature, both from government and social entrepreneurship perspectives, regarding exogenous events and pandemics in general and COVID-19 literature in particular, from the perspective of the context-responsiveness framework for the relationship between government and social entrepreneurship [21]. The current research contributes to this framework by explaining the interdependence of government and social entrepreneurial endeavours between 2010 and 2020 through the lens of the COVID-19 public health emergency.

This study is built around a thorough review of the literature about the nature, scope, theoretical foundations, and contributions of social entrepreneurship in both developing and developed societies to show how important social entrepreneurship is in times of crises and disaster. Then, with the current situation in mind, a mind-mapping tool was used to come up with themes and subthemes about how the government helps or hinders social entrepreneurship. In later sections, a detailed discussion is presented regarding the themes generated through content analysis. The paper concludes with a discussion, implications for theory and practice, and limitations.

## 2. Literature Review

### 2.1. Theoretical Basis of Social Entrepreneurship

Many definitions of social entrepreneurship exist in the literature, and each of them agree that their prime motive is to benefit society. Merging this definition with the entrepreneurship description, social entrepreneurs are also considered to be constantly innovative and creative to achieve their goals; specifically, they are referred to as “entrepreneurs with a mission”, “catalysts for social transformation”, and “social problem solvers”. There is a need for social entrepreneurship because the government has a limited budget, leaving some social services to be provided by social entrepreneurs through social innovation [23,24,25,26].

Similarly, Austin et al. (2012) made an important contribution in this regard by contrasting commercial entrepreneurship on one end of the spectrum with social entrepreneurship on the other and establishing that both of these notions are intertwined, that is, social or charitable work has economic support and may generate cash, whereas economic endeavours generate some social value in the name of corporate social responsibility and employer branding [27]. Smith (1981) also mentions the intertwining of these two concepts, claiming that commercial entrepreneurs, while acting in their private interest to make a profit, produce goods and services that the public wants at a price they can afford or are willing to pay, and thus act in a socially desirable manner [28]. However, in terms of difference, these researchers contend that social entrepreneurship is undertaken when there is a social need unmet by commercial entrepreneurs and businesses.

Further, in terms of the theoretical underpinnings of the nexus between social entrepreneurship and government, a context-responsiveness framework is discussed by Erpf et al. (2019) [21], stating that social entrepreneurship has a greater chance of survival with favourable environmental characteristics and a dynamic relationship between the organization and the environment [29]. In addition, an entrepreneur’s success does not depend solely on his or her traits; the infrastructure, including the social entrepreneurial organizations and the environment created by the government, is equally important since these organizations are under the authority and scrutiny of the government bodies that have administrative powers.

### 2.2. Social Entrepreneurship and Its Relation with Social Work and Charity

Social entrepreneurship has surfaced as a rapidly growing area of inquiry in terms of research/theory and practice and has been pondered through the lenses of business and management, non-profit institutions, public policy, and healthcare. However, not everyone agrees on how to define it or how important it is to an economy [30]. Nevertheless, the management field views it as an organizational form merging commercial logic with social welfare [31].

Concerning the difference between social work and charity, it has been pointed out that social work is the most basic kind of service. It meets the needs that social and business entrepreneurs cannot because they cannot reach the places where the service needs to be performed. Thus, when social entrepreneurship starts to cater to the previously unmet or ignored need(s), the role of social work recedes. Thus, they have a complementary yet mutually exclusive relationship.

In addition, charity, social work, and social entrepreneurship all have blurry lines between them, but charity or philanthropy can be a motive for social entrepreneurship. For example, the Karachi-based Hamdard Foundation is a good example of a social enterprise that combines social work and social entrepreneurship. The Hamdard Foundation was the first to offer herbal and inexpensive medicine to the general public. In addition, the foundation gave money to the largest public university in Karachi and helped people in Pakistan, India, and Bangladesh when there was a natural disaster or other crisis [32]. However, in the case of charity, the sole motive is quick salvation rather than finding a long-term solution to a problem due to the despondency of the less privileged [24].

### 2.3. Disaster Management through Social Entrepreneurship

Public health emergencies and disasters, whether natural (earthquakes, famines, and tsunamis) or man-made (wars, terrorism, and human activity-induced public health emergencies such as spill-out of toxic chemicals), negatively impact not only the physical lives of people in terms of injury, death, destruction, and loss of property and physical spaces but also market spaces (such as stores) and places of social connections (such as places of worship). Public health emergencies and disasters can happen at anytime and anywhere [33]. According to Newton (1997), catastrophes are not secluded incidents; rather, they are a societal phenomenon that occurs inside a social system [34]. This loss of physical and social places is first remedied by social entrepreneurs and then by commercial entrepreneurs.

Social entrepreneurs are very important before, during, and after disasters because they act as a buffer to soften the blow and fight for their communities through activism and lobbying. For instance, before the onset of a natural disaster, they inform the community about its magnitude, scope, and likelihood of occurrence. In addition, they assist in the evacuation of susceptible locations and the establishment of makeshift residents.

In times of emergencies, misinformation creates panic [35], and social entrepreneurs during these times help with relief, give verifiable information about current losses, and make accurate estimates of the damage that could happen in the future. In addition to supplying people with food, shelter, water, and basic medical care, these groups coordinate volunteer efforts to find missing individuals, provide aid and assistance, and advocate for the allocation of government resources for public services such as the return and rebuilding of their communities [36].

### 2.4. Social Entrepreneurship in Developed versus Developing Countries during Disasters

In the last two decades alone, the world has witnessed many natural and human-induced disasters. The most notorious natural disaster includes the 2004 Tsunami that hit densely populated regions around the Indian Ocean, devouring approximately 310,000 people, with Indonesia being the most affected country. Similarly, in 2005, a deadly earthquake jolted Pakistan’s region of Kashmir, killing more than 100,000 people from the developing country. In the same year, a series of hurricanes struck the Gulf States of the United States, taking more than 2000 precious lives and causing losses of approximately USD 250 billion [37].

Studies of developing countries show that disasters such as those listed above hurt the economy by lowering GDP, making trade worse, and causing budget imbalances that may be hard to fix in the long run. Developed countries, on the other hand, perform well in disasters and can handle the effects on their own because they have more resources. However, disasters have different effects on different parts of the same country, with poorer areas performing worse than wealthier areas [37]. Social entrepreneurship is not a construct of the financial affluence of a nation, since even middle- to low-income countries utilise a higher percentage of their GDP for charitable and other social causes. Asian World magazine says that religiosity is a better way to tell which countries perform more charitable work, since 78% of social work and charitable donations by Muslims occur during Ramadan [38]. The studies showed that the complex structures and response frameworks in the institutions of developing countries increase the number of opportunities for social entrepreneurship [39]. Social entrepreneurship is often observed to emerge in countries where the capacity of national governments to address social issues [40], particularly those concerning public health emergencies, is limited.

### 2.5. How Social Entrepreneurs Are Responding to COVID-19 with Business Guidelines

Even during the pandemic and other disasters when social entrepreneurship is most needed, it is not an easy job. Organizational issues and societal consensus are two serious issues, except for other hurdles [41,42]. These problems can lead to inconsistency in service delivery when it is needed the most because of the absence of a continuity plan [43]. The COVID-19 pandemic asserted the need for social entrepreneurs with innovative initiatives to facilitate the vulnerable segments of society.

Unsurprisingly, social entrepreneurs responded accordingly and came up with many innovative solutions, products, and services while maintaining the mandatory social distance. On the one hand, these innovative ideas lessen loneliness by providing connectivity and social services, and on the other hand, they share the burden of already over-occupied government institutes around the world. Several examples are provided below.

ConnectHear interpreted the COVID-19-related news and guidelines in sign language for people with hearing issues (hailstone 2020). Nanoclo designed a facemask with bacteria-resistant nanofibers; this mask decreased the price of n95 masks significantly. Renowned couture designers and entrepreneurs collaborated to create less expensive and more standardised PPEs (personal protective equipment) for health professionals [44,45]. SehatKahani.com (accessed on 5 October 2022) and Corona92.com (accessed on 5 October 2022) were ICT-enabled solutions based on qualified home-based female doctors and reliable and authentic statistical analyses regarding the COVID-19 outbreak, respectively [46,47].

Fundación Capital, a south American social enterprise mainly focused on monetary aid to the less privileged with the help of ICT, aligned its chatbots to disseminate useful information about preventive measures, government directions, and socio-economic well-being through managing finance in COVID-19-like emergencies. On the other side of the world, in North Africa, a Jordanian initiative, in collaboration with a pharmaceutical firm and the Ministry of Health, established a hotline for the people of the MENA region to obtain health-related consultation from certified medical practitioners. In time of disasters and pandemics, an integrated healthcare system is necessary to properly tackle the emergency [48].

Dimagi, an American entrepreneurial tech firm, offered CommCare, a mobile data platform for all pandemic response efforts such as case management, surveillance, diagnosis data management, and contact tracking. The least developed region of Sub-Saharan Africa also experienced the same enthusiasm from social entrepreneurs, where Simprints offered the same kind of services through tech aid.

Quarantined at home, whether from the outside world or with a potential abuser, is a dreadful fate. The SCHSA (Senior Citizen Home Safety Association) in Hong Kong stepped up to provide companionship and emergency assistance to the city’s elderly and others in need around the clock in the wake of isolation caused by the COVID-19 pandemic. To help with the latter, CrisisTextLine provides instant, secure, and anonymous online access to certified counsellors via short message service (SMS) messages. This programme operates nonstop in the United States, the United Kingdom, and Canada, with a particular emphasis on cases of child abuse and domestic violence.

Enterprise resilience, which is essentially the capacity to continue operating during times of crises and upheaval, is a crucial attribute [49]. This capability is supported by the presence of pre-crisis resources that are used in a strategic way to lessen the effects of the crisis [50]. Similarly, this enterprising tenacity is essential throughout the COVID-19 health emergency. Little research on crisis management in the context of entrepreneurship explores the efforts entrepreneurs take to mitigate the severity of the ramifications, such as sales, marketing, management of human resources, and employment practice adjustments. Small businesses run by entrepreneurs are more flexible and are able to change [51], and the same was expected from them during COVID-19 and the dynamics that followed it [19]. Entrepreneurial crisis management is strongly related to Mallak’s (1998) concept of “bricolage,” in which instead of adopting inflexible practices during COVID-19, iterative and adaptable techniques using effective reasoning are adopted [52]. In addition, Martinelli et al. (2018) found that resilient entrepreneurs are those who use their limited resources to create positive social change (change and opportunity). In this time of need, the aforementioned enterprises and initiatives around the world have exhibited precisely this quality [52].

Moreover, it is important to consider the threat to the sustainability of social entrepreneurship activities due to problems related to the coronavirus, such as social distancing and lockdowns. During COVID-19, social entrepreneurs faced unique challenges not seen in previous crises: social distancing while providing essential services. The critical challenge for social enterprises was survival and how they were going to serve people during this crisis, which might have lasted for an unidentified time, and it is not always possible or feasible for social entrepreneurs to provide essential and required services to the intended communities while maintaining social and, more importantly, physical distancing. Even though many commercial entrepreneurs are remodelling their businesses to accommodate social distancing, the same can be difficult for social entrepreneurs due to the nature of the service they provide to disadvantaged communities, such as the rapid provision of food and medicines while maintaining the prescribed physical distance, whereas adapting to a virtual model of social enterprise takes time and does not convey the full array of intended benefits to the target segment of society [53].

### 2.6. Paradox of Social Entrepreneurship at the Time of Social Distancing during COVID-19

By March 2020, the WHO reported about 0.4 million confirmed cases and nearly 20,000 deaths globally, leaving a devastating impact in 197 countries [54]. The only method for controlling it was to interrupt, limit, or eliminate physical proximity between people (person-to-person physical contact) for a predetermined period, which was designated as quarantine, social distancing, and isolation, respectively [55].

Governments across the globe launched a public awareness campaign in conjunction with a prolonged lockdown to improve the chance of social distancing at maximum. Physical distancing was a feasible and life-saving solution for preventing the virus’s spread and for relieving pressure on the affected countries’ already frail healthcare systems [56]. Nevertheless, research indicates that epidemics in general and social isolation in particular might trigger post-traumatic stress symptoms [57,58]. In addition to emotional concerns, irritation, and loneliness, social separation has caused substantial physical inconveniences, such as limited food and resources.

This has opened a door and given social entrepreneurs a big chance to make sure that the physical, emotional, and financial needs of the general public and the unprivileged are met.

Even though academic research on post-disaster recovery often looks at commercial entrepreneurship, there is not much written about the important role social entrepreneurs play in helping communities rebuild and recover during and after disasters. In addition, there is no mention of the challenges that social entrepreneurs have to face to improve the socioeconomic health of the public by expanding their community-oriented services or starting new ones to meet the increased needs of those affected by disasters and calamities, such as foodstuff, makeshift residence, or monetary, emotional, and spiritual support [36].

However, despite its importance, social entrepreneurship is not without challenges, the majority of which are imposed by the government, which stifles the progress of a social entrepreneur through bureaucracy, causing delays in the provision of services or the rebuilding of communities. Moreover, hurdles are faced due to the government being either under-cautious of the looming disaster (easily detected type 1 error) or acting too cautiously and risk-averse, which hampers its decision-making efficiency (hard-to-notice type 2 error). These acts by the government divert social entrepreneurs from socially productive activities to finding creative ways to navigate bureaucratic obstacles. The solution to these and related problems include removing or reducing artificial barriers to entry for social and even commercial entrepreneurs, reducing bureaucratic hurdles by realizing that the government policies pre-disaster are not necessarily relevant for during and post disaster and hence need revision, and reducing unnecessary government intervention in the operations of social entrepreneurs.

In the context of global pandemics, social entrepreneurship is very important, but the current stream of COVID-19-related studies does not discuss how social entrepreneurship can be started and continued in a creative and efficient way despite the limits of social distance, in order to help communities and make them more resilient. This study aims to fill that gap.

## 3. Data and Analysis

### 3.1. Planning the Literature Search for Analysis

To ensure rigor and replicability for this study, a systematic literature review was conducted to provide a comprehensive summary of all existing research on the given topic. The goal of this approach was to find, review, and combine relevant studies in a way that is clear and easy to repeat. This review process began with the establishment of rules and limitations for an extensive literature search with the intent of evaluating and classifying the raw data ontologically [59,60]. The principles of precision, coverage, transparency, and exhaustive synthesis were upheld [61].

### 3.2. Conducting Literature Search for Analysis

In the first step, the boundary of the “social entrepreneurship” term and initial search criteria were adopted from Mair and Martí (2006) [62]. This concept entails creating value for communities and society through innovative resource amalgamation, restructuring, and deployment. Then, the intention of the innovative exploitation and exploration of resources is not commercial or profit-based. In terms of process, it involves providing much-requested services and ideas for smaller organizations. The expansive body of literature not only provided guidance for the search of key terms in the selected search forums but also set inclusion and exclusion criteria for the current study.

Books and book chapters were excluded since the COVID-19 pandemic is a current pandemic, and books on social entrepreneurship from the perspective of this pandemic are scarce and can potentially suffer from an incoherent peer review process and/or limited access. On the other hand, it is commonly accepted that journal articles represent a validated form of knowledge, as they undergo a rigorous peer-review process before publication and are often considered authoritative sources in academic fields [63]. Notwithstanding, the main focus of the current study is on reports and magazine articles since the aforementioned source of knowledge is not sufficient for in-depth systematic analysis as opposed to magazines, reports and news article, which are updated frequently regarding the significance of social entrepreneurship pre-, post- and during disasters and pandemics, including but not limited to COVID-19. In this way, all journal and magazine articles that have been published are accessible to anyone and meet the entry requirements. This method is advantageous for the latest research ideas that are still in the development stage. It also makes it easy to replicate and extend [59].

The range of the search was between 2010 and 2020, both inclusive; 2010 was chosen as the starting point due to the epidemic of SARS in the same year. The search was initiated with terms extracted from the relevant literature. This query retrieved relevant titles from the Google Scholar search engine. To widen the search, each researcher looked at the whole study and compared it to the criteria for inclusion and exclusion. The resulting list was compared to a set of criteria to identify missing elements. This thorough process led to the categorical data of 67 publications and reports about the different problems and opportunities that social entrepreneurs face during and after pandemics and natural disasters. The search did not include social firms that did not address the entrepreneurial components. There are exceptions for articles discussing the early stages of social enterprises.

The government’s role in encouraging or discouraging social agents during and after epidemics, natural catastrophes, and other such external events was identified using manifest content analysis, an inductive method. The research concentrated on a typical government function during epidemics. Following this, post-disaster and post-pandemic conditions were analysed to identify potential openings for aspiring business owners. When considering their activities in the present and the future, entrepreneurs can benefit from using this method to create a timeline of how they can respond to various events and public health emergencies [64]. (Please see Appendix A for literature search procedure)

### 3.3. Conducting the Analysis

As an alternative to the deductive method, we use an inductive approach to theme identification by employing an ontological and thematic procedure (Appendix B) [65]. Consequently, the study’s structure and nature were defined by its identification and classification. The research problem and hypotheses in this study are grounded in the statements that were identified as part of the studies that met the inclusion criteria, and the themes that emerged from the analysis of those statements represent the central ideas, perspectives, and theoretical connections among them [61,66]. On the dataset, qualitative thematic coding was performed. Primary and secondary themes were better understood with the research goals, theoretical foundation, and methods.

In contrast to traditional content analysis, which extracts themes from decontextualised content, this research extracts themes from reports and articles through rigorous searching and thorough comprehension [67]. Therefore, the names of the subjects were taken from the existing corpus of literature. Iterative, exhaustive, and thorough steps were taken to discover and confirm the themes. In the first round, many polished ideas came to the surface. After that, we put them into groups and used them to move the domain’s ontological structure forward. The classification process echoes the context of the study and it is flexible [68]. Following a thorough examination for redundancy and repetition at each level, the data were organised vertically following general ontological design principles, such as the creation of a distinct superclass above the subclass and similar class.

Two themes were evaluated based on scope ontology, which is a way of putting together similar first-order themes. After arranging them in vertical order, second-order themes were placed together into a thematic area. This process kept going until two main areas of social entrepreneurship research were established. Government support for social enterprise (SE) during and after a disaster or pandemic (Class I) and governmental restrictions on SE during and after a disaster or pandemic (Class II) were identified as the two main categories from the 67 studies. A thematic map detailing the ontological frameworks of social entrepreneurship was created in MindMeister.com (accessed on 8 January 2023) (Figure 1). The primary appendages branching out from the left side of the map show the difficulties, potential rewards, and government participation in social entrepreneurship during epidemics. It stretched to third-order themes. They are specialised subsets that elaborate on the potential risks that the government presents to social entrepreneurs during and after pandemics. This process repeated itself until a total of 70 studies was compiled, at which point they were assessed and examined in terms of the challenges and opportunities facing social entrepreneurs ((Jones et al., 2011) were followed for research design and methodology).

## 4. Results and Discussion

The ontology of social entrepreneurship is shown in a thematic map (Figure 1), which graphically identifies the organization of the findings and conclusions. The results of the study show that after a pandemic or natural disaster, social entrepreneurship is most likely to be affected by opportunities, problems, and government policies.

Literature about government policies regarding social entrepreneurship discusses the impact of government actions on the workings of social entrepreneurship. Analysis of the data found two thematic areas in the literature, focusing on (1) supportive government policies/actions and (2) challenging government policies/actions.

There are two types of government policies. Those initiatives that promote social entrepreneurship such as financial assistance [68,69] business insurance [70,71], local body support and initiatives for coordination between stockholders for broadening social impact [72,73] and those that promote government-sponsored training for entrepreneurs [74,75]. There is ample evidence that entrepreneurial training supports improving age-appropriate competencies of entrepreneurial alertness and efficacy. However, in order to produce the best results, entrepreneurship education and training must be tailored to the regional context.

### 4.1. Government Policies Inhibiting Social Entrepreneurship

Challenges comprise the first second-order theme in the thematic area of government policies. These studies discuss the challenges that social entrepreneurs can and have to face from red tape, bureaucracy, formalities, and stern regulations. Bureaucratic hurdles and post-disaster regulations emerged as the second-order theme in this thematic area. According to the research, complicated rules and procedures, non-supportive behaviour, and the regulatory framework for post-disaster activities pose significant challenges for social entrepreneurs [76,77,78]. However, there is sufficient evidence available that establishes the government’s roles at different levels, both as an inhibitor and a catalyst of social entrepreneurship [79].

Ayiro (2010), in his study about the critical role of social entrepreneurship in AIDS management in education institutes in Kenya, concluded that despite the ineffective policies of the government regarding the spread of AIDS in schools, the government is not willing to blur the boundaries between social entrepreneurs and government departments so that they can collectively curb the spread of the disease through the introduction of innovative approaches and the extension of market principles [75].

Similarly, Dhesi (2010) talked about how the diaspora has a stake in the building, development, and progress of their ancestral village because they have an emotional connection to it. They ran into many problems as they tried to reach their goal and carry out social projects [7]. Among them, the most severe were imposed by the indifferent behaviour of the local bureaucracy and the conflict between formal local institutions such as the village council and slow-adjusting, culturally embedded informal institutions. In addition, local institutions are often seen as broken, and elected leaders cannot work towards sustainable development goals because they do not have the moral authority to do so. This leads to slow progress in social entrepreneurship at the grassroots level, and they try to overcome it by operating through non-governmental organisations (NGOs).

Finally, Bonnici and Raja (2020) quoted an example of Village-Reach (public health provider), which is a social venture in Mozambique, to establish that social endeavours face funding crises and that the challenge is to win the government’s attention, interest, and funding [80]. This case shows how hard it can be to work with the government on large-scale projects to make systemic changes. It also shows how hard it can be to use money from international donors to put new practices into government service delivery.

### 4.2. Government Policies Promoting Social Entrepreneurship

Government support is the second first-order theme in the thematic area of government policies. Articles and reports on this theme correspond to seven different first-order themes: financial assistance by the government, favourable policies to cover and support the entrepreneurial initiative, regulations by the government for insurance coverage of not only entrepreneurs but also of entrepreneurship in case of any undesirable circumstances, support by the local authorities, and an initiative by the authorities to call and invite stakeholders, including entrepreneurs, for a social cause in the times of pandemics, and the last first order theme is the government-sponsored specialised and hands-on training for capacity enhancement of entrepreneurs.

The most important and talked about topic is the government’s financial support for social entrepreneurs. The themes improve the understanding of the sources of financing for social entrepreneurs in different regions around the world and report that the majority of social entrepreneurs employ personal resources and funds. Studies also report two important sources of funding, including entrepreneurs’ family banks and crowd funding [68,69]. The basic challenge for social entrepreneurship emerges because the priority is on social goals instead of financial ones, and it does not meet the interests of traditional forms of finance [77]. Montgomery et al. (2012) described this situation as one in which social entrepreneurship is collaborative and collective and consequently draws on support and alliances from multiple actors, especially the government, to build a venture [65]. One more theme that emerged in the area of government support is favourable policies for social entrepreneurship. This emphasises how regulatory bodies’ encouraging policies and initiatives, as well as an inspiring environment, can act as catalysts for social entrepreneurs to grow in difficult times for the mutual benefit of society [81,82,83].

A considerably more important second-order theme that appears as a parallel to local government support is the role of the local body or local government. First, these studies show how important it is for the two stakeholders to work together since they both want to solve problems in society [73]. Further in this theme, studies have discussed the elements of hybridity, that is, how social entrepreneurs address social issues that the government fails to capture [31], yet they face the counterproductive activism of the government’s actions [84]. The government support theme asserts that activities of social entrepreneurship are mostly dependent on government regulations and multiple support measures, including the normative and cultural cognitive components [69,84]. Smeets (2017) discovered the reasons for the differences between the two, despite having the same end goals, as differences in procedures and organisational logic [85]. There is a need to simplify this complex relationship through involvement, a consolidated timeframe, and clearly defined support mechanisms to establish a conducive environment for social entrepreneurship [86,87]. Studies assert that local authorities play this pivotal role since they help in the configuration of the ecosystem of social ventures and since they facilitate social entrepreneurship in conjunction with social services such as healthcare, education, and welfare.

Finally, researchers on government support issues have included two more domains: engaging or inviting stakeholders for social causes and government-sponsored training. One second-order theme that emerged here is initiatives for coordination between stockholders that tend to facilitate social entrepreneurship. Collective efforts through collaboration with multiple stakeholders facilitate the leveraging of existing resources and the moulding of institutional arrangements and policies to bring a positive transformation to society. Through the examination of several instances of collective social entrepreneurial pursuits, Shockley and Frank (2011) explored the role of government for multi-institutional and multi-stakeholder involvement in bringing societal betterment during and after disasters [40]. This engagement helped social entrepreneurs sort out their current needs and assess the required resources. Broadening the scope of this area, Montgomery et al. (2012) and Trivedi (2010) explored the dynamics of collective social entrepreneurship, including what it entails and what kinds of skills and competencies are required for new social change enthusiasts and volunteers in a collective level to bring about social change, especially during and after disasters [68,88]. This led to another second-order theme, which is entrepreneurial training programs. These trainings are designed for social change, sponsored by the government and conducted by social entrepreneurs, to enhance the skills of new aspirant entrepreneurs and social entrepreneurs, as well as of people who volunteer their services for recovery and rehabilitation to operations [74,75]. Official engagement by the government in different services recognises the existence and need of social enterprises that lead to feelings of support and encouragement.

Conclusively, it is said that governments around the world at different levels have recognised the potential of social entrepreneurs to fulfil the social and economic agenda, and it is now a fact that social entrepreneurs can and are bridging the gap between social services and government [88,89].

## 5. Conclusions

There is a ubiquity of social and environmental issues around the globe. As a result, there is a call for politicians, business leaders, and members of civil society, the government, and semi-government organizations to focus their efforts on these goals. Even so, there are not clear lines between the issues the government handles and the social and environmental problems that can be left to the market and other non-government institutions to handle in part or wholly. To sum up, the difference between how much people want essential services and how much formal and informal institutions can give them is an opportunity for social entrepreneurs.

This research aimed to present a state-of-the-art systematic literature review of social entrepreneurship from the perspective of disasters and pandemics generally and COVID-19 specifically. The databases employed for the keyword search included ScienceDirect and Google Scholar. This step led to the shortlisting of 67 reports, journals, and magazine articles, and content analysis through NVivo software generated themes in the process. These themes included challenges and support from the government.

This study concludes that there is a need to relax government regulations for social entrepreneurship, especially during and after public health emergencies of greater magnitude. These revisions should be able to alleviate the burdensome obstacles caused by the bureaucracy. This study found evidence that, besides the financial support of different types and scales, the scope of social entrepreneurial training should be widened to improve relevant skills and expertise.

Furthermore, policymakers must recognise the contributions of commercial and, in particular, social entrepreneurs during disasters and recovery. The former strengthens the communities by facilitating the provision of essential goods and services to the affected population and thereby generating employment and other benefits, and the latter creates social transformation directly by creating societal resilience and making the transition from pre-disaster life to post-disaster life successful. A social entrepreneur can come from the community, business, or religious institutions, civic activists, scholars, or reformers, and is not limited to non-governmental organizations. Thus, it is important to provide a conducive environment for both types of entrepreneurs to step in during times of crisis since they are in a better position to assess local situations and public needs in times of rapid changes post-disaster and since they work on the grassroots level and reduce signalling noise or conflicting policies by the government [90].

This research contributes to the existing body of knowledge by catering to the social entrepreneurship literature in the context of pandemics and disasters, including the COVID-19 pandemic, through pertinent examples and evidence from the developing and developed parts of the world. The second contribution of this study lies in the underpinning of collaboration between government and social entrepreneurship to track the all-inclusive scenario in the difficult times of social distancing, ranging from opportunities provided by the government to the challenges posed by the government in furthering the cause of social entrepreneurial endeavours, especially in difficult times of public health emergencies.

Albeit any predicament faced by social entrepreneurs, in times of public health emergencies such as COVID-19, the world needs social entrepreneurs the most that have social innovations to address unprecedented social and societal problems. Social entrepreneurs have a deep sense of how to empower communities and always respond proactively to the immediate needs of the people they serve. In the next section, the opportunities that the coronavirus will bring in contrast to its devastation will be analysed, because during the coronavirus crisis and after the end of the pandemic, there will be an entrepreneurial boom [91].

The limitations of this study are twofold. Firstly, there is the potential lack of comprehensive and up-to-date information on government policies, programs, and initiatives that may have impacted social entrepreneurship during 2010–2020. Even though the study conducted a thorough review of available literature and data sources, there may have been important government actions or decisions that were not captured in the analysis. Secondly, the available literature focuses on developed and less developed countries, with limited coverage of the least developed and lowest income countries. This could limit the generalizability of the findings to other contexts. As a result, the study may not fully capture the complexity and diversity of government support and hurdles for social entrepreneurship during the studied period.

## Figures and Tables

**Figure 1 ijerph-20-05071-f001:**
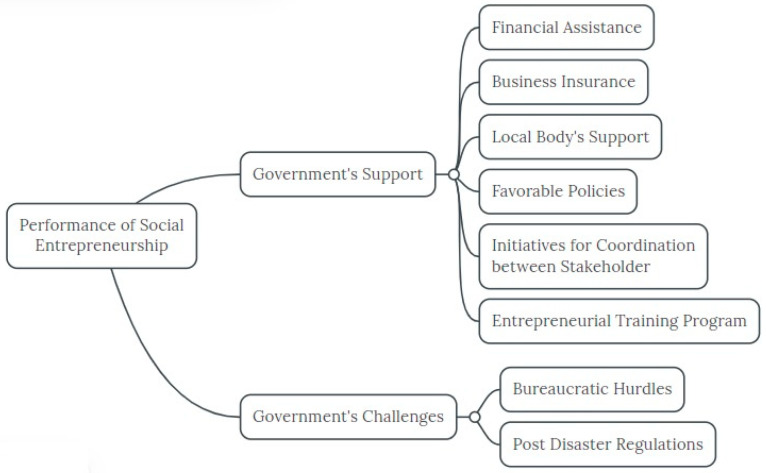
Thematic map of government support and challenges to social entrepreneurship.

## Data Availability

Data are publicly available in shape of research studies and articles and references.

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
