# Peer review of "Government as a Facilitator versus Inhibitor of Social Entrepreneurship in Times of Public Health Emergencies"

_ijerph, 2023, doi:10.3390/ijerph20065071_

Round 1

Reviewer 1 Report

Dear authors: 

They have done a good job. I would like to make some comments: It is advisable to use data such as WOS, SCOPUS to reinforce the quality of the bibliographic review. I would like to know what are the boolean connectors used.
In the text, the authors reference appendix C, but I can't find it. The bibliographic review provides the reader with an overview of the state of the art in social entrepreneurship during times of crisis.

Yours faithfully

Author Response

Dear Reviewer,

I would like to express my gratitude for your valuable contribution to the review process of our research article. Your critical insights and constructive feedback have helped us to improve the quality and accuracy of our work significantly.

Your meticulous examination of the manuscript, thoughtful comments, and suggestions for improvement have been invaluable in refining the content and methodology of our study.

We appreciate your time, effort, and dedication in providing a thorough and insightful review of our work. Your feedback has been crucial in shaping the final version of our article, and we are incredibly grateful for your contributions.

Point wise responses to your suggestions/comments/observations are as follow

Sincerely,

  • “appendix C” was typed mistakenly, deleted in the manuscript now.
  • I have added some work from the suggested databases, however, this study focuses on the government’s role in public health emergency of COVID-19, this phenomenon is not old and there is comparatively less work available about this issue. That is why reports and articles from less know generals and forums were included in the study.
  • Boolean Connectors “ Year: 2010-2022 Title, abstract, keywords: "Government AND Social entrepreneurship"

Reviewer 2 Report

The structure of the paper should be revised. The title of sections is not supported by the content of the section.

Results and discussion not linked with the previous sections. 

Author Response

Dear Reviewer,
I would like to express my gratitude for your valuable contribution to the review process of our research article. Your critical insights and constructive feedback have helped us to improve the quality and accuracy of our work significantly.
Your meticulous examination of the manuscript, thoughtful comments, and suggestions for improvement have been invaluable in refining the content and methodology of our study.
We appreciate your time, effort, and dedication in providing a thorough and insightful review of our work. Your feedback has been crucial in shaping the final version of our article, and we are incredibly grateful for your contributions.
Point wise responses to your suggestions/comments/observations are as follow;

Sincerely,

  • The introduction has been enhanced with some additions and deletions to better frame the study and its objectives.
  • While some new literature has been included, the study found limited research on government responses to public health emergencies during the period of 2010-2020, which was the focus of the study.
  • The design and structure of the study have been improved, with content rearranged into more appropriate sections and headings added for clarity.
  • The methodology section has been revised with numbered headings and more detailed information provided in the appendix about the research process and steps.
  • The results section has been enhanced with additional information and figures to better illustrate the findings.
  • The conclusion has been enriched with more content and suggestions for future research.
  • The new structure of the study is now more coherent and clear, with a stronger link established between the results and previous sections.

Reviewer 3 Report

Overall it is interesting and well designed study with a lot of information but the problem of this paper is the poor organization of individual sections.

There is a major error in citing the references in the manuscript which is confusing the reader. This means that the manuscript have not been prepared and reviewed well before submitting it. It is a common mistake when changing the manuscript format from publisher to another, but it changes the manuscript totally. The citation should be cited in mdpi joural format.

Section 2/3 I think part of included information should be separated and add to new section METHODS. Otherwise you can combined both section to one.

Also, the information from appendix (i.e. criteria and others) should be added to method section to describe how this process was conducted.

Results should be separated from discussion

I do not see any limitations of this research.

A short, introductory paragraph summarizing the intent and scope of the study would provide a useful context for the rest of the paper. In addition, a closing paragraph or two with summative insights, or overarching principles garnered from this review would make the manuscript more complete. As written, the final paragraphs seem an abrupt end to the work and the paper lacks a sense of closure.

Some recommendations to be included in text

https://doi.org/10.3390/ijerph18115888

https://doi.org/10.3390/su12208561

https://doi.org/10.3390/su13084517

Author Response

Dear Reviewer,

I would like to express my gratitude for your valuable contribution to the review process of our research article. Your critical insights and constructive feedback have helped us to improve the quality and accuracy of our work significantly.

Your meticulous examination of the manuscript, thoughtful comments, and suggestions for improvement have been invaluable in refining the content and methodology of our study.

We appreciate your time, effort, and dedication in providing a thorough and insightful review of our work. Your feedback has been crucial in shaping the final version of our article, and we are incredibly grateful for your contributions.

Point wise responses to your suggestions/comments/observations are as follow;

Sincerely,

  • Individual sections have been revised.
  • Though the sentences have been rephrased to sync with the overall body of the paragraph, if the mistake is still there, please highlight a few to help me please.
  • The citation style would be revised in accordance with the MDPI format at the production stage.
  • Contents of the sections have been moved to more appropriate sections, it really helped to make the study more clear and coherent
  • If you allow it and if it doesn’t lower the quality of the study, the information in appendices will serve better in this form, if I present it in descriptive form, it will make it wordier with unnecessary elaboration.
  • Limitations have been added at the end of the conclusion section
  • A final paragraph has been added at the end of the conclusion to end the research with a sense of completion.
  • Thanks for providing the so relevant and updated studies, it really helped me to understand the public health emergency. All three studies have been cited.

Round 2

Reviewer 3 Report

I'm glad to hear that my feedback was helpful in improving the quality and accuracy of your research article. It's great to see that you took my suggestions seriously and made the necessary revisions to make the study more clear and coherent. I appreciate your efforts in revising the individual sections.

I hope that the final version of the article is successful and contributes to the understanding and prevention of public health emergencies.